# Highly Species-Specific Foliar Metabolomes of Diverse Woody Species and Relationships with the Leaf Economics Spectrum

**DOI:** 10.3390/cells10030644

**Published:** 2021-03-13

**Authors:** Rabea Schweiger, Eva Castells, Luca Da Sois, Jordi Martínez-Vilalta, Caroline Müller

**Affiliations:** 1Department of Chemical Ecology, Bielefeld University, 33615 Bielefeld, Germany; rabea.schweiger@uni-bielefeld.de; 2Departament de Farmacologia, Terapèutica i Toxicologia, University Autònoma de Barcelona, Cerdanyola del Vallès, 08193 Catalonia, Spain; eva.castells@uab.cat; 3CREAF, Cerdanyola del Vallès, 08193 Catalonia, Spain; l.dasois@creaf.uab.cat (L.D.S.); Jordi.Martinez.Vilalta@uab.cat (J.M.-V.); 4Departament de Biologia Animal, Vegetal i Ecologia, University Autònoma de Barcelona, Cerdanyola del Vallès, 08193 Catalonia, Spain

**Keywords:** chemodiversity, deciduous versus evergreen, leaf economics spectrum, leaf habit, Mediterranean, metabolomics, metabolite richness, primary metabolites, specialized metabolites, species comparison

## Abstract

Plants show an extraordinary diversity in chemical composition and are characterized by different functional traits. However, relationships between the foliar primary and specialized metabolism in terms of metabolite numbers and composition as well as links with the leaf economics spectrum have rarely been explored. We investigated these relationships in leaves of 20 woody species from the Mediterranean region grown as saplings in a common garden, using a comparative ecometabolomics approach that included (semi-)polar primary and specialized metabolites. Our analyses revealed significant positive correlations between both the numbers and relative composition of primary and specialized metabolites. The leaf metabolomes were highly species-specific but in addition showed some phylogenetic imprints. Moreover, metabolomes of deciduous species were distinct from those of evergreens. Significant relationships were found between the primary metabolome and nitrogen content and carbon/nitrogen ratio, important traits of the leaf economics spectrum, ranging from acquisitive (mostly deciduous) to conservative (evergreen) leaves. A comprehensive understanding of various leaf traits and their coordination in different plant species may facilitate our understanding of plant functioning in ecosystems. Chemodiversity is thereby an important component of biodiversity.

## 1. Introduction

Plants produce an astonishing diversity of organic molecules in terms of biosynthetic origin, structure and function. More than one million metabolites of low molecular weight may occur across all plant species [1]. The primary metabolites, e.g., sugars, organic acids and amino acids, are essential for maintaining cellular homeostasis and are involved in growth, development and reproduction. Probably less than 10,000 primary metabolites exist, which are found to be more or less ubiquitous in all plants [2]. In contrast, the much more diverse specialized (or secondary) metabolites play major roles in interactions of plants with the abiotic and biotic environment and are specific for certain taxa [3,4,5]. The biosynthetic pathways for primary and specialized metabolites are closely interlinked, as specialized metabolites are synthesized from the primary ones [6,7]. Diversification in primary metabolites thus contributes to a high chemodiversity of specialized metabolites [7], while also other mechanisms reinforced the diversification of the specialized metabolism during evolution [4,8]. However, whether primary and specialized metabolites correlate in numbers and whether their composition is interlinked in plants has rarely been addressed.

The metabolome of a given individual is highly complex and a result of gene by environment interactions. The genetic repertoire largely determines which metabolic pathways are expressed in a species, with some specialized metabolites occurring in various taxa due to convergent evolution of biosynthetic pathways [2]. Moreover, taxa differ in their chemodiversity such as the metabolite richness, i.e., the number of metabolites. These differences across taxa have been, for example, shown for (semi-)polar metabolites in leaves [9] and root exudates [10]. When comparing the chemical composition among species, their phylogenetic relatedness should be taken into account, similarly as is done in comprehensive biodiversity studies [11,12]. Phylogenetic imprints in the metabolome are a result of past and current environmental factors that shape the metabolic composition of plants [4], leading to species-specific metabolomic niches [13]. Ecometabolomics tools are increasingly used to explore such taxon-related differences but also to uncover species-specific responses to certain environmental conditions [9,14,15,16]. Common garden studies are a useful approach to compare traits across organisms under standardized conditions, minimizing environmental variation.

In addition to species-specific metabolic phenotypes, plant species differ in a range of morphological, physiological and phenological functional traits, i.e., traits that affect individual fitness indirectly via their impacts on performance [17]. A well-known concept that summarizes resource use strategies in plants is the leaf economics spectrum, which describes a continuum from conservative leaves, characterized by slow returns on investment of nutrients and carbon (C), to acquisitive leaves, having the opposite properties [18,19]. Acquisitive leaves tend to show shorter lifespans and higher specific leaf areas (SLA), mass-based nitrogen contents (N_mass_) and photosynthetic capacities (A_mass_) [18]. Leaf habits are hence closely related to the leaf economics spectrum, with deciduousness being associated with acquisitive leaves and evergreenness with conservative ones. These different leaf strategies should be mirrored in foliar metabolism, as metabolic pathways are closely linked to leaf growth [20], photosynthesis [21] and allocation of resources [13,22]. Thus, distinct metabolomes for species of different functional types (i.e., deciduous versus evergreen) as well as links between the primary and specialized metabolome and leaf SLA, C and N can be expected. Relationships between chemical and morphological traits across a range of plant species have been rarely investigated (but see, e.g., [20,23,24]). Disentangling these relationships may improve our understanding of how plants adapt to the various challenges of climate change [25].

In this study, we aimed to explore the composition of primary and specialized metabolites in the leaves of 20 woody species co-occurring in the Mediterranean region when grown under common garden conditions as saplings, using a comparative ecometabolomics approach. Mediterranean forests and woodlands show a high biodiversity in woody species but are also threatened by different components of global change [26,27]. Comprehensive knowledge of various leaf traits of these plant species and their coordination may facilitate efforts to enhance the resilience of these ecosystems. Specifically, our objectives were to (i) assess the chemodiversity (metabolite richness) and metabolic composition of leaves across species in relation to potential phylogenetic imprints, (ii) test whether primary and specialized metabolite richness and metabolic patterns are correlated and (iii) investigate relationships between foliar metabolomes and leaf resource use strategies (i.e., the leaf economics spectrum), including leaf habit and resource allocation patterns depicted as SLA, C and N stoichiometry. We argue that plant chemodiversity should be integrated into ecological studies as being an essential part of plant biodiversity that can offer intriguing insights in plant strategies to adapt to changing environments.

## 2. Materials and Methods

### 2.1. Plant Species

We selected 20 woody plant species that are representative of the Mediterranean vegetation and cover a wide range of leaf resource-use strategies. Half of the species are deciduous, whereas the other species are evergreen. In addition to two gymnosperm species of the Pinaceae, 18 angiosperms belonging to 11 families were included (Table 1). The leaf habits are distributed across taxa and from three families (Fagaceae, Oleaceae, Adoxaceae) both deciduous and evergreen species were chosen.

### 2.2. Experimental Design and Plant Sampling

Saplings of the 20 woody species (24 individuals per species) were grown in a common garden at the experimental fields of IRTA at Torre Marimon (Caldes de Montbui; 41.613 N, 2.170 E, 176 m a.s.l.), 30 km north of Barcelona (Catalonia, Spain). The climate is Mediterranean with warm and dry summers and mild winters, with average temperatures ranging from 5 °C to 27 °C and an average annual precipitation of 633 mm. Saplings (2–3 years old) were purchased from a tree nursery (Vivers Carex, Cornellà del Terri, Spain) to ensure homogeneity in size and growth conditions. In December 2018, individual plants were transplanted to 40 L pots placed on top of upside down plastic trays to keep the pots 5 cm above the ground for better drainage and to avoid direct contact with the soil. Moreover, pots were covered near the base of the stems with mulch to protect the soil from direct sunlight radiation, thus reducing growth of herbs, avoiding warming effects and reducing evaporation. Each pot was filled with 30 L of substrate and a 10 L layer of gravel below the substrate to ensure water draining. The substrate (pH = 7) was composed of 23% sand and 77% of a mixture of two commercial substrates for Mediterranean plants: BVU substrate (Burés, Girona, Spain) containing fertilizer NPK 15-7-15, *Sphagnum* peat and ground pine bark as well as J-2 substrate (Burés) containing compost, *Sphagnum* peat and perlite. Pots were randomly arranged in a ca 1000 m^2^ field and subjected to the local environmental conditions including precipitation. Additionally, plants were individually irrigated to field capacity by an automatic dripping system. In early May 2019, pesticides were applied once on *Populus alba* (Confidor, Bayer, Barcelona, Spain) and on *Pistacia lentiscus*, *Salix cinerea* and *Quercus ilex* (Breaker Max, Certis, Alicante, Spain) due to the presence of insect herbivores. In late May 2019, we selected six individuals per species (a total of 120 individuals) that showed no signs of damage by herbivores or pathogens. For each individual we sampled the second youngest expanded leaf or leaves of one or several branches corresponding to growth of the current season until enough plant material was obtained for chemical analyses (at least 100 mg fresh mass). Leaves were immediately frozen in liquid nitrogen, lyophilized and thoroughly ground. Additional leaves of similar age were sampled from the same plant individuals to determine the specific leaf area (SLA). Fresh leaves were scanned and areas were estimated using ImageJ software [28]. Leaves were then dried at 70 °C for 48 h, weighed and SLA was calculated as the ratio of leaf area to leaf dry mass.

### 2.3. Chemical Analyses of Leaves

Profiling of polar metabolites, mostly including primary metabolites (e.g., sugars, organic acids and polyalcohols), followed a modified procedure according to Schweiger et al. [9]. Leaf material was extracted with chloroform, methanol and water (1:2.5:1, *v*:*v*:*v*) containing tartaric acid (99%; Panreac, Castellar del Vallès, Spain) as internal standard, including extensive vortexing. The chloroform phase was discarded to reduce interference of non-polar metabolites with the target metabolites. Aliquots of the aqueous phases were dried in a rotary evaporator and metabolites were subsequently derivatized at 37 °C by methoximation with methylhydroxylamine hydrochloride (>97%; Sigma Aldrich-Merck, Madrid, Spain) and silylation with *N*-methyl-*N*-trimethylsilyltrifluoroacetamide (>98%, Sigma Aldrich-Merck) for 90 and 30 min, respectively. The samples were analyzed by gas chromatography coupled with a flame ionization detector (GC-FID, Agilent Technologies 7820A; Santa Clara, CA, USA) using a VF-5 ms capillary column (Agilent, 30 m × 0.25 mm × 0.25 μm) and a helium flow of 1.2 mL min^−1^. Samples were injected in pulsed-split mode (1:10) and run with the following temperature program: initial temperature 80 °C held for 3 min, ramp 5 °C min^−1^, final temperature 325 °C held for 3 min. One representative sample per species was analyzed by GC coupled to mass spectrometry (GC-MS, Agilent Technologies 7890A, equipped with an insert MSD with triple-axis detector 5975C) using the same column and temperature program as described for GC-FID. Kováts retention indices (RI) were determined [29] by comparison with a set of n-alkanes (C9-C36; Restek, Bellefonte, PA, USA) and peaks were identified by their RI and mass spectra in comparison with commercial standards (Sigma and Panreac). Blank samples containing the derivatization reagents were injected in the GC-FID and GC-MS every 12–15 plant samples.

Metabolic fingerprinting of (semi-)polar molecules of low-molecular-weight, mostly including specialized metabolites, was performed according to Schweiger et al. [30] with some modifications. Leaf samples were extracted with ice-cold 90% (*v:v*) methanol for 15 min in an ultrasonic bath. The extraction solvent contained one of two internal standards: hydrocortisone (>98%; Sigma-Aldrich, Steinheim, Germany) for gymnosperms and mefenamic acid (Sigma) for angiosperms, respectively. Different internal standards were used to avoid co-elution with plant metabolites, as revealed in pre-tests with all species. Extracts were filtered through 0.2 µm filters (Phenomenex, Torrance, CA, USA) and samples were analyzed using an ultra-high performance liquid chromatograph coupled to a quadrupole time-of-flight mass spectrometer (UHPLC-QTOF-MS/MS; UHPLC: Dionex UltiMate 3000, Thermo Fisher Scientific, San José, CA, USA; QTOF: compact, Bruker Daltonics, Bremen, Germany). Metabolites were separated on a Kinetex XB-C18 column (150 × 2.1 mm, 1.7 µm, with guard column; Phenomenex) at 45 °C and a flow rate of 0.5 mL min^−1^ using a gradient from eluent A, i.e., Millipore-H_2_O with 0.1% formic acid (FA), to eluent B (acetonitrile with 0.1% FA): 2 to 30% B within 20 min, increase to 75 % B within 9 min, followed by column cleaning and equilibration. The QTOF was operated in negative electrospray ionization mode. Centroid data were taken at a spectra rate of 5 Hz in the *m*/*z* (mass-to-charge) range of 50−1300. The settings for the MS mode were: end plate offset 500 V, capillary voltage 3000 V, nebulizer (N_2_) pressure 3 bar, dry gas (N_2_; 275 °C) flow 12 L min^−1^, low mass 90 *m*/*z*, quadrupole ion energy 4 eV, collision energy 7 eV. Measurements in MS mode were used for quantification (see below); for features that exceeded a certain intensity threshold, MS/MS spectra were obtained in addition in a separate chromatogram trace via the AutoMSMS mode using N_2_ as collision gas and applying *m*/*z*-dependent ramping of isolation widths and collision energies. For recalibration of the *m*/*z* axis, a calibration solution containing sodium formate was introduced into the system prior to each sample. Four separate blanks for angiosperms and gymnosperms were measured under the same conditions.

Foliar C and N contents were determined using a C/N analyzer (Vario MICRO Cube, Elementar Analysensysteme, Hanau, Germany). Data were expressed both on a per-mass (C_mass_, N_mass_) and on a per-area (C_area_, N_area_) scale. Only those samples were retained in the overall dataset for which all data (GC-FID, UHPLC-QTOF-MS/MS, SLA, C and N) were available; five samples had to be discarded due to a lack of sample material or because of technical issues with some of the measurements, thus reducing the sample sizes of some species (*PH*, *BS*, *AM*, *FA*, *FL*) to n = 5.

### 2.4. Data Processing and Statistical Analyses

The analytes measured by GC-FID were quantified via their peak areas using the OpenLab EZChrom edition (Agilent). The relative retention time (RT) for each analyte from the GC-FID analyses was determined using the internal standard as a reference and data were then aligned using the R package *GCalignR* [31]. Peaks present in three or more blank samples and those present in less than half of the plant samples within all species were removed from the final dataset in order to retain only those compounds that were representative for a species. For metabolites that formed different analytes during derivatization (i.e., fructose, glucose, galactose), the peak areas of the analytes were added. Concentrations were expressed relative to the peak areas of the internal standard and sample dry mass.

The UHPLC-QTOF-MS/MS data were processed in DataAnalysis v4.4 (Bruker Daltonics). After recalibration of the *m*/*z* axis for each sample, metabolic features (each characterized by a specific *m*/*z* at a certain RT) were picked and quantified via their peak heights with the Find Molecular Features algorithm using spectral background subtraction and the following settings: signal-to-noise ratio 3, correlation coefficient threshold 0.75, minimum compound length 20, smoothing width 5. Only features in the RT range 1.2–29 min were included, thus removing most primary metabolites co-eluting in the injection peak (<1.2 min). Features belonging to the same metabolite (i.e., isotopes, adducts, charge states, fragments based on a H_2_O loss) were sorted together in so-called buckets (also called compounds). From each bucket, only the feature with the highest intensity was further used. These features were aligned across samples using ProfileAnalysis v2.3 (Bruker Daltonics), allowing shifts of 0.1 min (RT) and 6 mDa (*m*/*z*). Then, peak heights were divided by those of hydrocortisone ([M+HCOOH-H]^−^ ion) and mefenamic acid ([M-H]^−^ ion) for gymnosperm and angiosperm species, respectively. Only those buckets were retained in the dataset for which the mean intensity in at least one species was more than 50 times higher than its mean intensity in the corresponding blanks, which occurred in at least half of the samples of at least one species, to focus on compounds representative for these species. Extracts of the used insecticides were measured under the same conditions and all features that occurred in these samples were removed from the dataset. Peak intensities were divided by the sample dry mass.

The term ‘metabolic feature’ is used for all entities measured, i.e., analytes and identified metabolites (GC-FID) as well as metabolic features (*m*/*z* at a certain RT; UHPLC-QTOF-MS/MS). For the sake of simplicity, hereafter we call ‘primary metabolites’ the features that were measured via GC-FID and ‘specialized metabolites’ those measured via UHPLC-QTOF-MS/MS, because these are the main targets for each method, although we acknowledge that other metabolites may also be present in either method.

Statistical analyses were performed in R 3.6.1 and R 3.6.2 [32] using different packages. Separate analyses were conducted for primary and specialized metabolites. To visualize sets of metabolic features occurring in certain plant species or combinations of species, UpSet plots were used (package *UpSetR*), showing only intersection sizes of at least two and 15 features for primary and specialized metabolites, respectively. A Pearson product-moment correlation test was used to test whether the metabolite richness (i.e., the number of metabolic features per sample) of primary and specialized metabolites correlated. To compare the metabolic composition between samples, non-metric multidimensional scaling (NMDS) analyses with Kulczynski distances were performed (package *vegan*), using two dimensions. For these analyses, feature concentrations were expressed on a relative scale, (i.e., a scale ranging from zero to one, with the sum of feature intensities being one for each sample) to account for the different internal standards that were used for gymnosperms and angiosperms in the UHPLC-QTOF-MS/MS measurements. Wisconsin double standardizations of square-root transformed data were applied. To test whether the primary metabolome and the specialized metabolome (both given as metabolic composition) correlated, a Mantel test based on a Spearman rank correlation and Kulczynski distances for both matrices (GC-FID and UHPLC-QTOF-MS/MS) was used. Different leaf traits were plotted as contour lines into the NMDS plots via the ordisurf function (package *vegan*) using generalized additive models (GAM) based on restricted maximum likelihood estimation with Gaussian error distributions, identity link functions and thin plate regression splines. In addition, linear mixed models were used to assess the relationships between NMDS axes scores (as explanatory variables) and leaf traits, including metabolite richnesses, SLA, C_mass_, C_area_, N_mass_, N_area_ and the C/N ratio. These models included species as a random factor to account for the fact that individuals within a species are likely to be more similar than across species. Models were fit and assessed using the R packages *dplyr*, *gamm4, ggplot2*, *Ime4*, *ImerTest*, *Matrix* and *MuMln*. Initially, the interactions between NMDS1 and NMDS2 were also tested, but were later on dropped from the models as they were not significant. Some traits (SLA, C_area_, N_area_, C/N ratio) were log-transformed to meet normality assumptions in linear models.

## 3. Results

We obtained 411 metabolic features for the primary metabolite analyses by GC-FID and 11,217 features (each represented by a *m*/*z* at a certain RT) for the specialized metabolite analyses by UHPLC-QTOF-MS/MS. The peak size for tartaric acid, the internal standard used for the GC-FID analyses, was slightly higher for the evergreen than for the deciduous species. The peak heights of the internal standards hydrocortisone and mefenamic acid (UHPLC-QTOF-MS/MS) were comparable across species, except for lower and more variable intensities of mefenamic acid in *A. campestre*. Circa one third of the primary metabolites (124; 30.2%) occurred in all 20 plant species (Figure 1a and Figure 2a), including all the metabolites that could be identified, namely four sugars (fructose, glucose, galactose, sucrose), two organic acids (oxalic acid, citric acid) and one polyalcohol (*myo*-inositol) (Appendix A). In contrast, only 17 (0.2%) of the specialized metabolites were found in all species and most of the specialized metabolites (5888; 52.5%) were found in one plant species only (Figure 1b and Figure 2b). In both datasets, we found indications for phylogenetic imprints, i.e., there were many features that exclusively occurred in the members of certain plant families (Figure 2). These phylogenetic imprints seem to be more pronounced for the specialized than for the primary metabolism. For example, the Salicaceae species *P. alba* (*PA*) and *S. cinerea* (*SC*) shared four features measured per GC-FID (i.e., 1.0% of the features measured with this analytical platform) that did not occur in any other species. For the specialized metabolites measured per UHPLC-QTOF-MS/MS, there were 351 (3.1% of the features measured with this platform), 323 (2.9%) and 121 (1.1%) features that exclusively occurred in the two species belonging to the Oleaceae, Pinaceae and Salicaceae species, respectively. The three *Quercus* species shared 165 (1.5%) taxon-specific features, and in addition there were 233 features (2.1%) exclusively shared by two of these oak species.

The total number of features per species was less diverse for primary than for specialized metabolites (Figure 2); it ranged for primary metabolites from 221 (*P. halepensis*) to 307 (*S. nigra*), while the number of specialized metabolites differed pronouncedly among species, being lowest in *B. sempervirens* (532 features) and highest in *B. pubescens* (2027 features). The number of features probably belonging to the specialized metabolism was, depending on the species, two- to eight-fold higher than the number of features assigned as primary metabolites. There was a significant positive correlation between the number of metabolic features of the primary metabolism and that of the specialized metabolism at the individual plant level (Pearson correlation including all species; *r* = 0.28, *p* = 0.003; Figure 3). This relationship became marginally significant (*p* = 0.065) when accounting for species effects using a linear mixed model.

The plant species showed distinct foliar metabolomes, as assessed by NMDS analyses (Figure 4; stress value for primary metabolites: 0.211; for specialized metabolites: 0.176). The differences between species were less pronounced for the primary than for the specialized metabolism (Figure 4a,b). The metabolic composition of species belonging to the same plant family was quite similar, indicating phylogenetic imprints. Species within these families partly overlapped for the primary metabolites (Figure 4a), whereas all species were well separated in the NMDS plot based on the specialized metabolites (Figure 4b). Although species differences were more pronounced for the specialized metabolites, the overall spatial arrangement of the species in the NMDS plots was comparable; indeed, the primary metabolome and the specialized metabolome were significantly correlated (Mantel test; r = 0.44, *p* < 0.001). The number of metabolic features per sample increased along the NMDS 1 axis for the primary metabolites, along with the mainly horizontal separation of the plant species (Figure 4a,c). For the specialized metabolites, a similar increase in the metabolite richness from the bottom left to the top right was observed (Figure 4d). These relationships were supported by the linear mixed models (Table 2).

The leaf metabolomes showed distinct patterns for deciduous and evergreen species for both primary and specialized metabolites; the deciduous species were clustered with the separation of the species according to leaf habits being more pronounced for the specialized than for the primary metabolites (Figure 4a,b). For the three plant families from which both deciduous and evergreen species were included (Fagaceae, Oleaceae, Adoxaceae), the deciduous ones clustered close to the deciduous species from other plant families, especially for the specialized metabolites (Figure 4b).

The differences in the metabolic composition between deciduous and evergreen species were related to the metabolite richness and to leaf traits linked to resource use strategies (Figure 4, Figure 5 and Figure 6 and Appendix A). The number of metabolic features was slightly higher for the deciduous than for the evergreen species. The leaf traits related to the leaf economics spectrum varied pronouncedly within and among species and leaf habits and the species rather formed a continuum along the trait value axes (Figure 4, Figure 5 and Figure 6 and Appendix A). As expected, most deciduous species had higher SLA, lower C/N ratios, lower C_area_ and higher N_mass_ compared to the evergreen species, except *B. sempervirens* (*BS*). Linear mixed models revealed significant effects of the composition of primary metabolites (both NMDS axes) on N_mass_ and C/N ratio and of the composition of specialized metabolites (only NMDS2) on C_mass_ and N_area_ (Table 2).

## 4. Discussion

Our comparative ecometabolomics approach revealed a high chemodiversity in the leaves of the 20 woody species grown as saplings under common garden conditions, with many more specialized than primary metabolites found per plant species. This finding fits well to the general view that specialized plant metabolites are much more diverse than primary ones [2]. We acknowledge that the chemodiversity in the investigated plant species is in fact much higher, as we only focused on the (semi-)polar metabolites, neglecting the less polar ones. The huge diversity of specialized metabolites results from various metabolic pathways, in which variable combinations of precursor subunits are used, as well as from multi-member gene and enzyme families, multi-product enzymes, low substrate specificities of certain enzymes and various modifications of backbones (e.g., methylation, glycosylation) within substance classes [4,33,34]. Across plant species, the metabolite richness differs pronouncedly, as shown in the present study and in other studies for herbs and trees [9,13,35]. We cannot rule out that the variability in the metabolite numbers is partly related to different leaf matrices affecting metabolite extraction and detection. However, the intensity of the internal standard for the primary metabolites was slightly higher for the evergreen species, which showed a lower number of metabolites, and no clear differences were found in intensities of the internal standards between deciduous and evergreen species for the specialized metabolites. Thus, structural differences of the leaves of distinct leaf habit probably did not largely affect the general pattern. More than half of all detected specialized metabolites were unique for one species, while less than 1% were shared by all 20 woody species investigated here. Almost the opposite pattern was found for the primary metabolites. Moreover, the leaf metabolic composition, which is based on both the presence and the concentrations of individual metabolites, largely differed between plant species, particularly for the specialized metabolites. This finding is consistent with other studies on leaf metabolomes across plant species [9,13,24]. Richness of both primary and specialized metabolites contributed much to the metabolic compositions observed, indicating that this measure of chemodiversity is already a very informative trait characterizing individuals and taxa. The leaf primary metabolism was not as universal and uniform as one may expect, as more than half of the detected metabolites were not shared by all 20 species and species were also separated according to their composition of primary metabolites, being mainly driven by differences in metabolite numbers. Likewise, in other studies several primary metabolites were found to occur only in some but not in other plant species (e.g., certain sugars or polyalcohols), while others are present in various species but in highly different abundances [9,20]. This indicates that different primary metabolites may have similar functions to maintain the cellular homeostasis and enable and shape growth and reproduction [20].

Furthermore, we found indications of phylogenetic imprints for the primary and particularly for the specialized leaf metabolites, as individuals belonging to the same species, and in some cases to the same families, shared many metabolic features and showed similar metabolic patterns. Such phylogenetic imprints, as well as species-specific metabolomes, also recently found for tropical tree species [13], may result from the ability of species to synthesize novel metabolites by gene duplication, followed by sub-/neofunctionalization and specialization of enzymes [34,36]. In addition, differences in gene expression can contribute to metabolic differences between species [4,33]. Compared to genes encoding enzymes of the primary metabolism, genes involved in the specialized metabolism are assumed to be more plastic due to less evolutionary constraints, allowing thus a high evolvability of specialized metabolites in a complex and changing environment [3,4,33]. Similar selection pressures in different taxa can lead to the evolution of distinct specialized metabolites with similar functions such as signaling, antioxidative or defense properties against herbivores and/or pathogens but also attraction of beneficial organisms [2]. Moreover, recent studies suggest that specialized metabolites play crucial roles in mitigating environmental stress, especially as an early response [37,38,39]. Overall, the species-specific diversity of the plant metabolome has been related with the functional diversity of species in the context of the ecological niche theory [13], highlighting the relevance of assessing chemodiversity across species in ecosystem-scale studies and considering it as a crucial component of biodiversity [40,41].

Across individuals of all 20 species investigated here, a significant correlation between primary and specialized metabolites regarding their numbers and patterns was revealed. This finding is coherent with the structure of plant metabolism, as primary metabolites serve as precursors and deliver energy for the synthesis of specialized metabolites [6,42]. A larger number of available precursors probably enables plants to produce more different specialized metabolites. Moreover, higher concentrations of certain precursors may allow enhanced metabolic fluxes from the primary metabolism to the specialized metabolism [20], because more building blocks and energy are available. In addition, the correlation may also indicate that primary and specialized metabolites work in coordination to fulfill certain functions in plants.

Furthermore, we could demonstrate that the foliar metabolomes of the woody species investigated here were partly related to other traits that are linked to the basic strategies of leaf resource use. The leaf economics spectrum provides a useful framework that explains general patterns of leaf resource use and investment on a continuous axis of variation through the coordination of leaf functional traits [18,43,44,45]. The species used in our study represent a continuum along the leaf economics spectrum, including some deciduous species with high SLA (e.g., *A. glutinosa*, *B. pubescens* or *S. cinerea*) towards the acquisitive side of the spectrum as well as some evergreens with low SLA (e.g., *P. sylvestris*, *B. sempervirens* or *P. lentiscus*) towards its conservative end. In accordance with the trait variation and trade-offs described by the leaf economics spectrum [18], the deciduous species showed higher SLA, lower C/N and a higher N_mass_ than the evergreen ones, suggesting that the deciduous species follow a more acquisitive resource-use strategy. Even within a plant genus, deciduous and evergreen species can co-exist in the same habitats, for example, the oak (*Quercus*) species investigated here and other oaks, which differ in certain leaf traits but not in others [46,47,48,49]. Likewise, although median trait values of the leaf economics spectrum differed between deciduous and evergreen species in our study, data largely varied with overlaps between groups, as reported before [19]. This variation may be due to within-group differences in leaf resource-use strategies, associated with varying lifespans and turnover rates of leaves or be related to species-specific adaptations to environmental factors that are not captured by the leaf economics spectrum. Moreover, as leaf development of evergreen trees may continue after leaves are fully expanded [50], differences between deciduous and evergreen species may be more pronounced later in the season. In deciduous species, much of the N is allocated to photosynthetic machineries, whereas in evergreens N is probably rather allocated to structural components [48]. Thus, the significant relationship between the primary metabolome and N_mass_ found in our study does not necessarily mean that there are more or higher concentrations of N-containing primary metabolites in leaves with high N_mass_. Photoassimilates derived from generally higher photosynthetic rates in deciduous species probably contribute to this finding. Further studies are needed to test the extent to which the investment into N-containing versus non-N-containing metabolites differs between deciduous and evergreen species. In any case, certain primary metabolites (e.g., sugars and organic acids) clearly affect leaf morphologies and physiological traits across species [20,21].

The potential relationships between plant chemistry and leaf resource-use strategies were raised decades ago by the carbon-nutrient balance [51] and resource availability balance [52] hypotheses. These hypotheses predict that slow-growing species adapted to low nutrient environments (potentially corresponding to the conservative leaf strategy) utilize C-based rather than N-based specialized metabolites and show high constitutive chemical defense levels compared to fast-growing species (i.e., acquisitive leaf strategy). In extensive studies devoted to test these hypotheses or their refined versions, including studies explicitly addressing the leaf economics spectrum [53,54], no general patterns of chemical variation according to plant life-history strategies associated with resource availability were found [55]. In the present study, patterns of primary and specialized metabolites tended to group species together by their leaf habit. Contrary to our findings, only weak differences in the chemical composition of leaves of woody deciduous versus evergreen species were found in other studies [53,56]. However, in 28 herbal species of the genus *Helianthus* the specialized metabolomes were correlated with a combination of traits related to the leaf economics spectrum [24]. Leaf metabolic traits and C/N stoichiometry also change with leaf age and plant development [57] and may respond to climatic differences between years in a species-specific manner, as shown for deciduous temperate trees [58]. Further studies are needed to explore such relationships across herbaceous and woody species with shared phylogenies and/or shared environments.

The adaptiveness of the species-specific combinations of traits of the leaf economics spectrum and the foliar metabolomes found in our study should be assessed in long-term studies in natural ecosystems. The traits associated with the leaf economics spectrum probably determine the performance of individual plants and species in environments of a given or changing nutrient availability [59,60]. Next to plant metabolites, leaf structural traits shape interactions of plants with their environment; for example, higher leaf toughness of evergreen species can reduce herbivory [54]. While specialized metabolites may be under a weak evolutionary constraint [4,61], the *combination* of different metabolites (i.e., the plant metabolomic niche) might be constrained by the plant life-history strategies. The leaf chemistry also affects leaf litter decomposition and nutrient cycling in ecosystems [58], which should be considered to fully assess impacts of leaf turnover and leaf quality on ecosystems. Further studies are needed to unravel how primary and specialized metabolites as well as leaf structure are affected by environmental challenges in the short-term and how they are interlinked and coordinated with functional adaptations in the long-term under diverse eco-evolutionary scenarios including ongoing climate change. In Mediterranean ecosystems, drought is a major stress factor, which should thus be taken into account when investigating traits and functions of species in these ecosystems.

## Figures and Tables

**Figure 1 cells-10-00644-f001:**
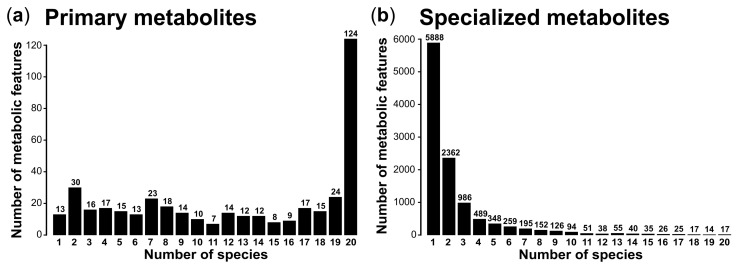
Frequency of occurrence of metabolic features of the (**a**) primary and (**b**) specialized metabolism detected in leaves of 20 plant species.

**Figure 2 cells-10-00644-f002:**
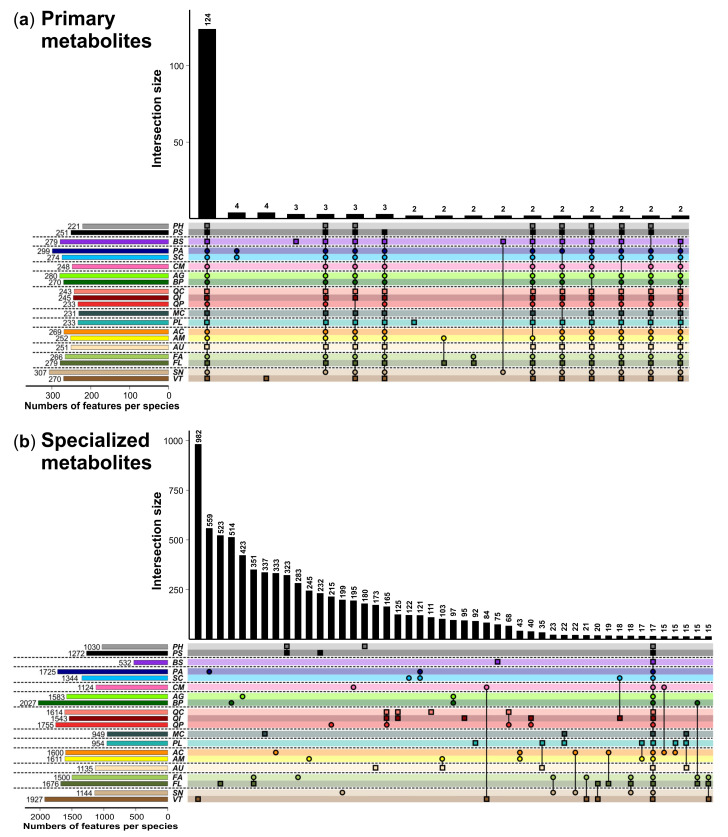
Total numbers (**left**) and occurrence of metabolic features of the (**a**) primary and (**b**) specialized metabolism detected in leaves of 20 plant species shown as UpSet plots (**right**). The vertical bar graphs show the number of features occurring in the (set of) species indicated with symbols below, with intersection sizes of at least two (primary metabolites) or 15 (specialized metabolites). Deciduous species are depicted as circles and evergreen species as squares. The horizontal dashed lines separate the species by plant families. For species abbreviations and taxon assignments, see Table 1.

**Figure 3 cells-10-00644-f003:**
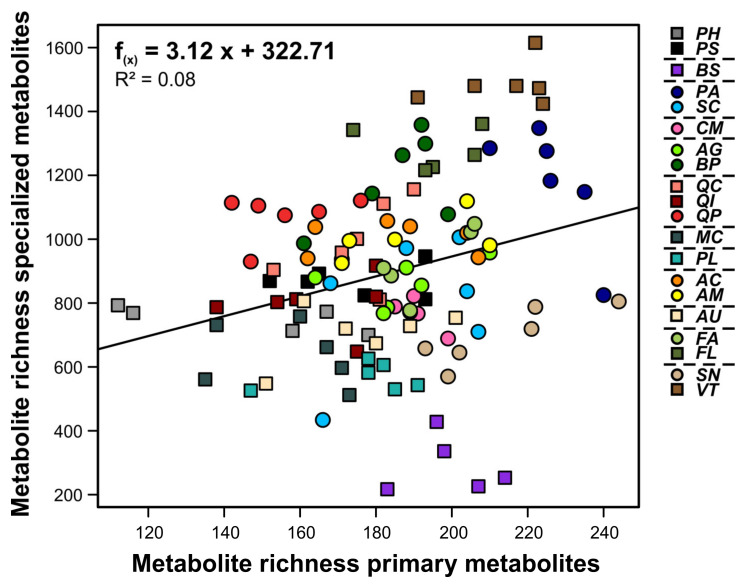
Relationship between numbers of metabolic features of the primary and specialized metabolism per individual detected in leaves of 20 plant species, with linear regression line. Deciduous species are depicted as circles and evergreen species as squares. The horizontal dashed lines in the legend separate the species by plant families. For species abbrevia-tions and taxon assignments, see Table 1.

**Figure 4 cells-10-00644-f004:**
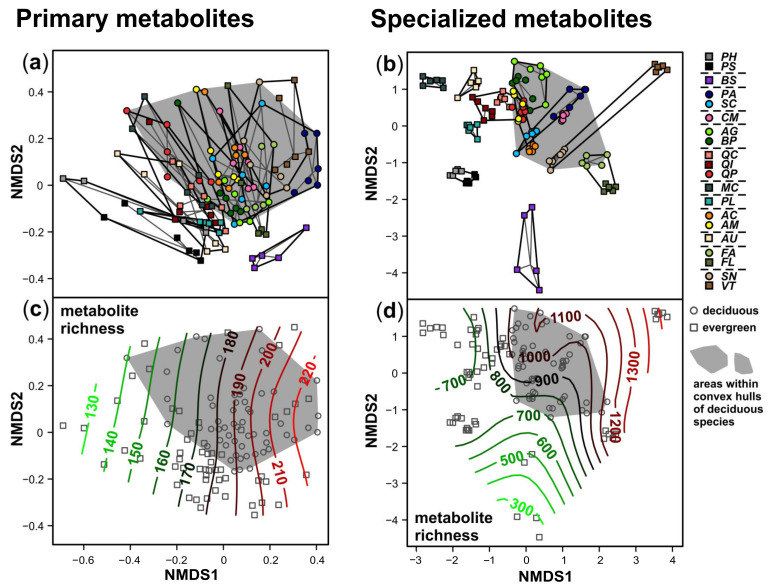
Non-metric multidimensional scaling plots of relative concentrations of metabolic features of the primary (**left**) and specialized (**right**) metabolism detected in leaves of 20 plant species (deciduous: circles; evergreen: squares). The deciduous species are highlighted by gray areas that are framed by convex hulls (closed curves surrounding all data points with minimum perimeter) for this group. The horizontal dashed lines in the legend separate the species by plant families. For species abbreviations and taxon assignments, see Table 1. (**a**,**b**) Data points are connected to the species medians (thin gray lines), families are surrounded by thick black lines. (**c**,**d**) Simplified plots with contour lines (green: low values; red: high) representing surface fits of generalized additive models for the metabolite richness of (**c**) primary and (**d**) specialized metabolic features.

**Figure 5 cells-10-00644-f005:**
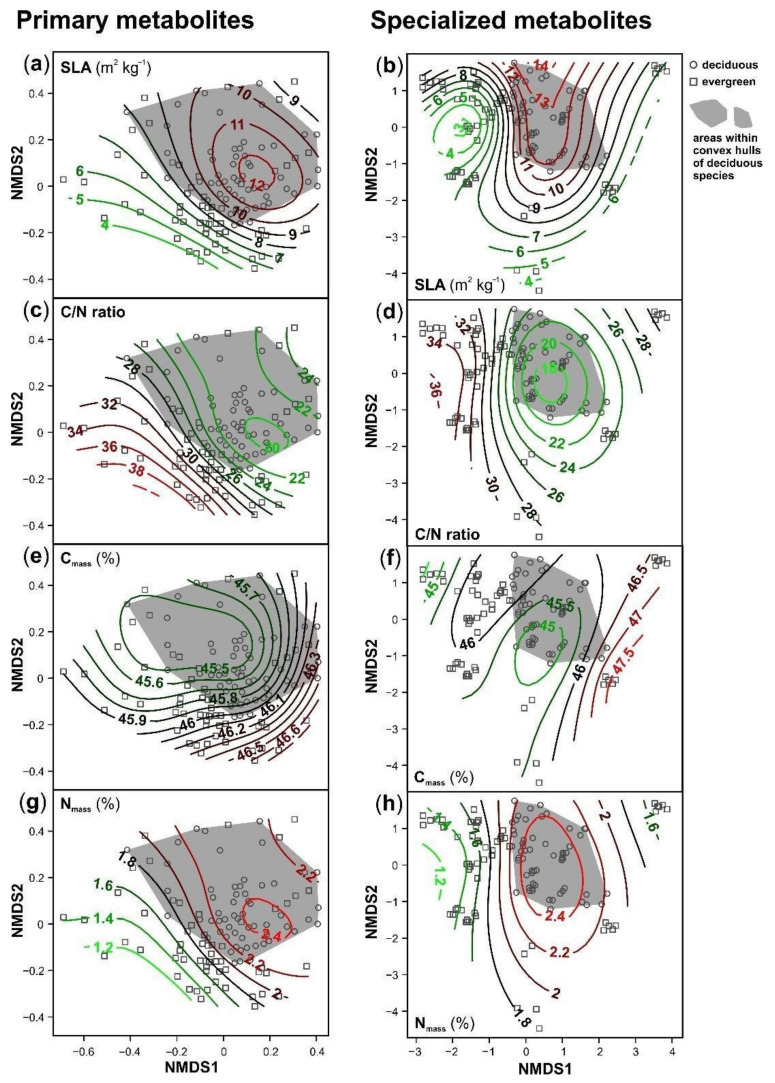
Simplified non-metric multidimensional scaling plots (for detailed plots see Figure 4) of relative concentrations of metabolic features of the primary (**left**) and specialized (**right**) metabolism detected in leaves of 20 plant species (deciduous: circles; evergreen: squares). The deciduous species are highlighted by gray areas that are framed by convex hulls (closed curves surrounding all data points with minimum perimeter) for this group. Contour lines (green: low values; red: high) representing surface fits of generalized additive models for: (**a**,**b**) specific leaf area (SLA); (**c**,**d**) carbon (C) to nitrogen (N) ratio; (**e**,**f**) C content (C_mass_); (**g**,**h**) N content (N_mass_).

**Figure 6 cells-10-00644-f006:**
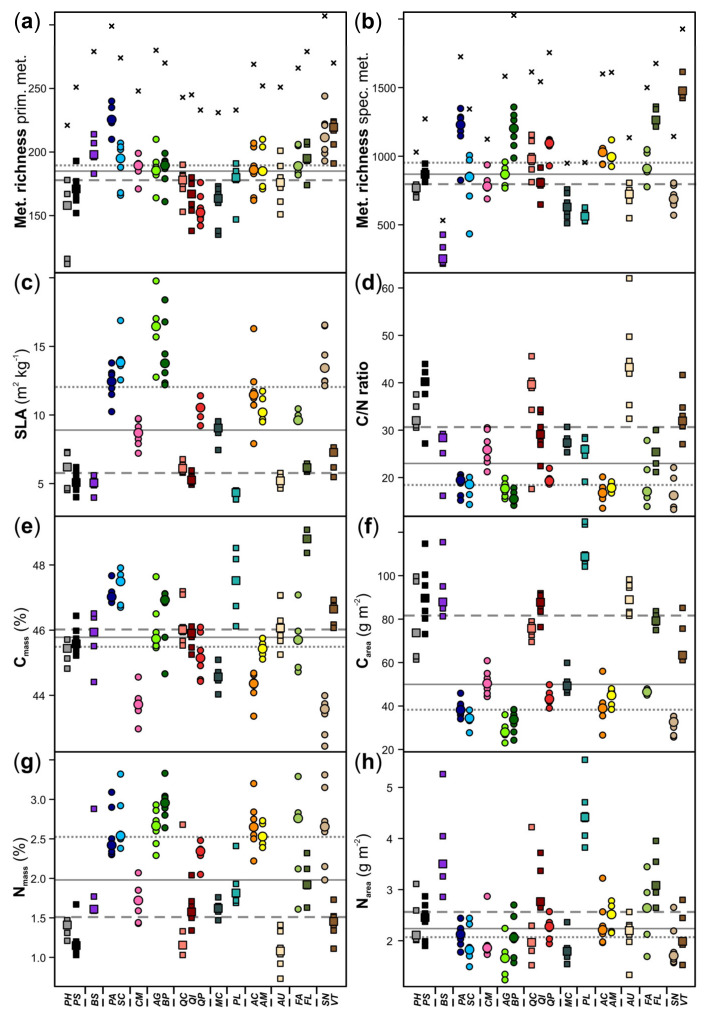
Stripcharts of leaf traits of 20 plant species (deciduous: circles; evergreen: squares): (**a**,**b**) metabolite richness of features of the primary and specialized metabolism; (**c**) specific leaf area (SLA); (**d**) carbon (C) to nitrogen (N) ratio; (**e**) C content (C_mass_); (**f**) C per leaf area (C_area_); (**g**) N content (N_mass_); (**h**) N per area (N_area_). Medians are shown as larger symbols (species) and horizontal gray lines (overall: solid; deciduous: dotted; evergreen: dashed), respectively. The vertical dashed lines at the x axes separate the species by plant families. For species abbreviations and taxon assignments, see Table 1.

**Table 1 cells-10-00644-t001:** Plant species used in the experiment, abbreviations (abbr.), taxa (classes, families) and leaf habits.

Species	Abbr.	Class	Family	Leaf Habit ^1^
*Pinus halepensis*	*PH*	Gymnospermae	Pinaceae	e
*Pinus sylvestris*	*PS*			e
*Buxus sempervirens*	*BS*	Angiospermae	Buxaceae	e
*Populus alba*	*PA*		Salicaceae	d
*Salix cinerea*	*SC*			d
*Crataegus monogyna*	*CM*		Rosaceae	d
*Alnus glutinosa*	*AG*		Betulaceae	d
*Betula pubescens*	*BP*			d
*Quercus coccifera*	*QC*		Fagaceae	e
*Quercus ilex*	*QI*			e
*Quercus petraea*	*QP*			d
*Myrtus communis*	*MC*		Myrtaceae	e
*Pistacia lentiscus*	*PL*		Anacardiaceae	e
*Acer campestre*	*AC*		Sapindaceae	d
*Acer monspessulanum*	*AM*			d
*Arbutus unedo*	*AU*		Ericaceae	e
*Fraxinus angustifolia*	*FA*		Oleaceae	d
*Phillyrea latifolia*	*FL*			e
*Sambucus nigra*	*SN*		Adoxaceae	d
*Viburnum tinus*	*VT*			e

^1^ Deciduous (d) or evergreen (e).

**Table 2 cells-10-00644-t002:** Estimates, standard errors and *p*-values of the linear mixed models used to test the relationships between non-metric multidimensional scaling (NMDS) axes scores (explanatory variables) and metabolite richnesses, SLA, C_mass_, C_area_, N_mass_, N_area_ and C/N ratio. Significant relationships (*p* < 0.05) are highlighted in bold.

	NMDS1			NMDS2		
	Estimate	Std. Error	*p*	Estimate	Std. Error	*p*
**Primary Metabolites**						
metabolite richness	103.229	5.920	**<0.001**	−8.323	5.838	0.157
SLA log (m^2^ kg^−1^)	0.145	0.126	0.255	0.081	0.102	0.425
C_mass_ (%)	0.465	0.587	0.430	0.295	0.481	0.541
C_area_ log (g m^−2^)	−0.136	0.130	0.298	−0.074	0.104	0.480
N_mass_ (%)	0.789	0.297	**0.009**	0.598	0.250	**0.018**
N_area_ log (g m^−2^)	0.034	0.164	0.835	0.015	0.142	0.917
C/N ratio log	−0.357	0.151	**0.020**	−0.268	0.126	**0.035**
**Specialized Metabolites**						
metabolite richness	173.650	25.300	**<0.001**	130.090	22.310	**<0.001**
SLA log (m^2^ kg^−1^)	0.029	0.046	0.527	0.015	0.038	0.703
C_mass_ (%)	0.048	0.176	0.788	0.388	0.161	**0.019**
C_area_ log (g m^−2^)	−0.029	0.047	0.537	−0.004	0.039	0.921
N_mass_ (%)	0.079	0.079	0.328	0.031	0.079	0.698
N_area_ log (g m^−2^)	−0.017	0.032	0.602	−0.076	0.035	**0.037**
C/N ratio log	−0.038	0.041	0.369	0.005	0.041	0.901

## Data Availability

The UHPLC-QTOF-MS/MS raw data (including MS and MS/MS data) and the bucket table will be published upon publication of the paper in the MetaboLights repository [62,63], being available with the accession number MTBLS2414 at www.ebi.ac.uk/metabolights/MTBLS2414 accessed on 15 January 2021. Further data presented in this study are available on request from the corresponding author.

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
