# Peer review of "Highly Species-Specific Foliar Metabolomes of Diverse Woody Species and Relationships with the Leaf Economics Spectrum"

_cells, 2021, doi:10.3390/cells10030644_

Round 1

Reviewer 1 Report

This is an informative comparative study examining differences among 20 woody species with respect to both primary and specialized metabolites. The metabolomic approaches are cutting edge and analyses were thorough. This is a substantial contribution as analyses of the metabolomes of wild plants is a new frontier, and should shed light on a very important aspect of plant phenotypes that, until recently, has not been possible. They showed that only 0.2% of the 11,000 specialized metabolites occurred in all species and most occurred in just one. In contrast, 30% of the primary metabolites occurred in all species. Particularly interesting results were that there was a positive correlation between the number of primary and specialized metabolites. Additionally, there was some variation among species in primary metabolism, which is generally considered to be quite conserved. Overall, I think this is a valuable contribution to an expanding field.

Clarifications on methods (section 2.4):

Because metabolomics is an emerging and evolving tool, I think it important to make the methods clear so other researchers can follow or assess their approach.

I think they are measuring ms/ms data for all compounds and NOT ms level 1. Can this be clarified. They are grouping features (mz/rt) into “buckets”, which I believe are analogous to compounds. On line 213, the authors state that they only kept a single feature of the highest intensity for each metabolite "bucket." If their data are ms/ms and not ms this could be a problem. If the dominant feature is commonly fragmented during ms/ms (i.e. gallate), this would lump different compounds that simply shared that feature. This means that their subsequent analysis of metabolic composition between samples only accounts for a single feature per compound ("bucket"), which could alter estimates of shared chemical composition. For example, it is possible that two "buckets" are both represented by some substructure (i.e. gallate) because this was the highest intensity feature for that retention time period. If two "buckets" characterized by the high intensity same substructure occur close to each other in retention time (within 6 seconds), they could potentially be characterized as the same compound, thus skewing the results to make it appear that two species containing these separate compounds would be more similar than they actually are. An alternative approach for grouping “buckets” is to use the COSINE score derived from uploading the entire ms/ms spectra for each compound (bucket) into GNPS. This avoids aligning compounds based solely on the tallest feature. It would be good to justify why they did not use the entire ms/ms spectra, and to acknowledge that they may be overestimating similarity among “buckets” and hence species.

The authors also remove any rare features that are not in at least half of the samples per species. We follow similar methods, but perhaps they could explain why they chose this specific cutoff.

Other comments:

Abstract:         The abstract could be more informative, eg what kind of correlation or relationship, positive or negative?

Line 103:         “In addition to two gymnosperm…”

Section 2.2:     The goal of the common garden was to eliminate environmental effects, yet plants were potted in Dec and leaves collected in May, only 5 months later. This is not a very long time to acclimate. Were the leaves collected for chemical analyses produced in these garden conditions?

Results:           They discuss phylogenetic imprint, but apparently this was determined visually from the figures. They should also use a statistical test. For example, a commone way to quantify phylogenetic signal is to use Blomberg’s K (Blomberg et al., 2003).

Figure 4 &5:    I was not familiar with interpreting the surface fits and the convex hulls. I think it would be helpful to have a more detailed description.

Author Response

This is an informative comparative study examining differences among 20 woody species with respect to both primary and specialized metabolites. The metabolomic approaches are cutting edge and analyses were thorough. This is a substantial contribution as analyses of the metabolomes of wild plants is a new frontier, and should shed light on a very important aspect of plant phenotypes that, until recently, has not been possible. They showed that only 0.2% of the 11,000 specialized metabolites occurred in all species and most occurred in just one. In contrast, 30% of the primary metabolites occurred in all species. Particularly interesting results were that there was a positive correlation between the number of primary and specialized metabolites. Additionally, there was some variation among species in primary metabolism, which is generally considered to be quite conserved. Overall, I think this is a valuable contribution to an expanding field.

REPLY: Thank you very much for this positive feedback.

Clarifications on methods (section 2.4):

Because metabolomics is an emerging and evolving tool, I think it important to make the methods clear so other researchers can follow or assess their approach.

REPLY: We agree that particularly for complex approaches that are still evolving like ecometabolomics, it is very important to describe the methods appropriately. Thank you for the specific suggestions to clarify our methods. We specified the indicated methods parts as described below.

I think they are measuring ms/ms data for all compounds and NOT ms level 1. Can this be clarified. They are grouping features (mz/rt) into “buckets”, which I believe are analogous to compounds. On line 213, the authors state that they only kept a single feature of the highest intensity for each metabolite "bucket." If their data are ms/ms and not ms this could be a problem. If the dominant feature is commonly fragmented during ms/ms (i.e. gallate), this would lump different compounds that simply shared that feature. This means that their subsequent analysis of metabolic composition between samples only accounts for a single feature per compound ("bucket"), which could alter estimates of shared chemical composition. For example, it is possible that two "buckets" are both represented by some substructure (i.e. gallate) because this was the highest intensity feature for that retention time period. If two "buckets" characterized by the high intensity same substructure occur close to each other in retention time (within 6 seconds), they could potentially be characterized as the same compound, thus skewing the results to make it appear that two species containing these separate compounds would be more similar than they actually are. An alternative approach for grouping “buckets” is to use the COSINE score derived from uploading the entire ms/ms spectra for each compound (bucket) into GNPS. This avoids aligning compounds based solely on the tallest feature. It would be good to justify why they did not use the entire ms/ms spectra, and to acknowledge that they may be overestimating similarity among “buckets” and hence species.

REPLY: We clarified this important point now. Yes, a bucket is a compound and this term is also used in the Bruker software for this; we clarified our wording throughout the manuscript by only using the words bucket(s)/compound(s) in the corresponding context. By using the so-called AutoMSMS mode of the UHPLC-QTOF-MS/MS, we measured both MS level 1 and MS/MS, but we only used the MS mode for quantification. Specifically, with the quite high spectra rate (5 Hz) we could quantify the metabolites at the MS level, whereas for some m/z (i. e., those exceeding a certain intensity threshold) we recorded MS/MS spectra in between, which were stored in a separate chromatogram trace. We clarified this now (lines 186-189). As we used only the MS level 1 for quantification, buckets contain MS features but not MS/MS fragment ions. Thus, we do not have the problem that co-eluting metabolites that differ in their precursor m/z but share a substructure that would produce the same MS/MS fragment are merged into one bucket. This can only happen if two metabolites co-elute AND have such a strong in-source fragmentation that even in MS mode they show a shared fragment that has a higher intensity than the precursor m/z; this is quite unlikely with the settings in the MS mode that we applied. Thus, we do not think that we overestimated similarities between species. The MS/MS data, which were not further used in our fingerprinting study but may be highly useful for the metabolomics community and hopefully contribute to the identification of metabolites of these and other non-model species in the future, will be made publicly available in the associated MetaboLights study, together with the MS 1 level data and the bucket table; we specified this now (line 519).

The authors also remove any rare features that are not in at least half of the samples per species. We follow similar methods, but perhaps they could explain why they chose this specific cutoff.

REPLY: The 50% cutoff-level is somehow arbitrarily chosen, but we thought that entities that occur so rarely that they are not characteristic, i. e., not occurring in at least half or in the majority of the sample for at least one species will not help in characterizing and comparing the species at the metabolic level. We included this explanation in the Methods now (lines 205-207, 227-228).

We excluded entities (analytes, identified metabolites, metabolic features) from the data sets that did not occur in at least half of the samples of at least one species. It is important to highlight here that we explicitly did not set any concentrations of the entities to zero in certain species only, as this would then suggest that the entities do not occur in these species, leading to wrong conclusions regarding which features are shared between species. As our peak detection algorithms (bot GC-FID and UHPLC-QTOF-MS/MS) validly separate peaks from background noise with the settings we applied, we are convinced that a metabolite really occurs in a species, even when found in only one sample.

Other comments:

Abstract:         The abstract could be more informative, eg what kind of correlation or relationship, positive or negative?

REPLY: We specified in the Abstract that both correlations were positive (line 22).

Line 103:         “In addition to two gymnosperm…”

REPLY: “Next to two gymnosperm…” replaced by “In addition to two gymnosperm…” (line 104).

Section 2.2:     The goal of the common garden was to eliminate environmental effects, yet plants were potted in Dec and leaves collected in May, only 5 months later. This is not a very long time to acclimate. Were the leaves collected for chemical analyses produced in these garden conditions?

REPLY: The common garden was set up to ensure homogeneous environmental conditions across all individuals and species, i.e. that each environmental effect would be identical for all individuals. We believe that plants had sufficient time to acclimate, as by the beginning of the experiment in May aerial parts of all individuals had new growth with no sign of stress (e.g. nutrient deficiencies). All leaves used for metabolomics analysis corresponded to new grown material produced in spring. We included this last information in the methods section (lines 135-136).

Results:           They discuss phylogenetic imprint, but apparently this was determined visually from the figures. They should also use a statistical test. For example, a commone way to quantify phylogenetic signal is to use Blomberg’s K (Blomberg et al., 2003).

REPLY: We assessed phylogenetic imprints/signals only visually and only at the family level, as we do not have genetic data for the plant individuals used in our study. A species-level phylogeny with published data may allow deeper insights related to the presence/absence of metabolites in species. However, we consider such a phylogeny less informative when assessing the metabolic compositions across species, as we partly found a huge metabolic variation within species, for which we could not account for with a species-level phylogeny. To better indicate that we did not statistically test for a phylogenetic signal, we rephrased several parts of the manuscript (lines 24, 282, 284, 417).

Figure 4 &5:    I was not familiar with interpreting the surface fits and the convex hulls. I think it would be helpful to have a more detailed description.

REPLY: We added some more detail in the legends: We now directly refer to the contour lines instead of surface fits (lines 359, 366-367) and explain what convex hulls are (lines 355-357, 364-366) (adjusted also in the supplement).

Reviewer 2 Report

This study reported that how the diversity of metabolic compounds differs among 20 wood species in relation to the leaf economics spectrum. I think this study has been well designed and conducted, the ms is well written. I have a few comments below. (Please note that I am a plant ecologist and not familiar with metabolomes, so I cannot examine the methods and results of the chemical analysis in detail.)

(1) My major concern is how the intractability itself affected the observed pattern in the diversity of compounds across species. Results of NMDS show the metabolic features were largely differentiated across the number of compounds (L383-384), and evergreen species tended to show a lower number of compounds (Fig. 4). Since evergreen species often have more defense or chemically recalcitrant compounds than deciduous species, I was a bit concerned that the NMDS patterns could be strongly affected by the extractability rather than the actual diversity of compounds. Since they used some internal standards when they extracted the chemicals, I suppose they could discuss this issue a bit more in the Discussion.

(2) The authors found a large overlap in the measured leaf traits between deciduous and evergreen species (L435-437). This was partly because they collected the leaf samples in May. It is well known that evergreen species developed leaves more slowly than deciduous species (i.e. leaf mass per area continues to increase for a few months after the full expansion), thus the difference in traits would be more pronounced when they compared the two groups in summer.

Other minor comments
L27-29 " We argue that a comprehensive understanding of various leaf traits and their coordination in different plant species may facilitate efforts to promote the resilience of ecosystems."
I think the authors did not discuss anything on this issue in the rest of the paper, so I think this text is relevant in the Abstract.

L 476-477 "However, leaf habit is not necessarily related to physical defenses [52]"
I think this text is not needed as it neither adds much to this paper nor to science. (I quickly checked the cited paper but I got a question whether they compared deciduous leaves and evergreen leaves at the right timing during the season. Also, they made mistakes in both terminology and units.) 

Author Response

This study reported that how the diversity of metabolic compounds differs among 20 wood species in relation to the leaf economics spectrum. I think this study has been well designed and conducted, the ms is well written. I have a few comments below. (Please note that I am a plant ecologist and not familiar with metabolomes, so I cannot examine the methods and results of the chemical analysis in detail.)

REPLY: Thank you very much for your valuable feedback. We included more details for some of the methods related to the chemical analyses now to better explain our methods and results to researchers who are less familiar with metabolomics approaches.

(1) My major concern is how the intractability itself affected the observed pattern in the diversity of compounds across species. Results of NMDS show the metabolic features were largely differentiated across the number of compounds (L383-384), and evergreen species tended to show a lower number of compounds (Fig. 4). Since evergreen species often have more defense or chemically recalcitrant compounds than deciduous species, I was a bit concerned that the NMDS patterns could be strongly affected by the extractability rather than the actual diversity of compounds. Since they used some internal standards when they extracted the chemicals, I suppose they could discuss this issue a bit more in the Discussion.

REPLY: We agree that for chemical analyses it is very important to optimize and standardize sample extraction efficiencies, especially when different sample matrices (e. g., different species as in our study) are included. Indeed, we cannot rule out that different extractabilities affected the metabolic diversity, measured as metabolite richness. We already acknowledged the limitations of our methodological procedure on extracting polar and semi-polar but not unpolar compounds in the first paragraph of the discussion of the submitted version of the manuscript. It is quite challenging to standardize sample extraction efficiencies across species, but we did our best to do so by thoroughly milling the dried leaves and extensively vortexing them. Additionally we applied ultrasonication for the samples measured by UHPLC-QTOF-MS/MS. Moreover, in the GC-FID samples we discarded the chloroform phase, thereby reducing matrix effects of non-polar compounds. As the internal standards were part of the extraction solvents given to the samples for extraction, we would expect lower peaks for these standards in those samples with a matrix hindering metabolite extraction and/or metabolite detection. In the case of tartaric acid, the internal standard used for the GC-FID analyses, the recovery was slightly higher in the evergreen species compared with the deciduous species. Furthermore, there were no obvious differences in the peak heights of mefenamic acid (i. e., the internal standard that was used for species of both leaf habits in the UHPLC-QTOF-MS/MS measurements) between deciduous and evergreen species; instead, one deciduous species (Acer campestre) showed lower mefenamic acid concentrations than the other species, pointing to some source of interference with the matrix during extraction and/or ion suppression during analysis for this species. Therefore, the lower number of metabolites found for the evergreen species do not appear to be influenced by a more complex matrix and extractability issues. We specified the Methods (lines 138, 147-148), added results regarding peak intensities of the internal standards (lines 271-275) and included a corresponding discussion point (lines 392-399).

(2) The authors found a large overlap in the measured leaf traits between deciduous and evergreen species (L435-437). This was partly because they collected the leaf samples in May. It is well known that evergreen species developed leaves more slowly than deciduous species (i.e. leaf mass per area continues to increase for a few months after the full expansion), thus the difference in traits would be more pronounced when they compared the two groups in summer.

REPLY: Thank you for pointing out this interesting discussion point. We only used young, fully expanded leaves. But, as you indicated, leaf development may have been still in progress, especially in the evergreen species. We included this aspect, together with a new reference (lines 467-469, new reference 50, subsequent numbers of references were adjusted).

Other minor comments
L27-29 " We argue that a comprehensive understanding of various leaf traits and their coordination in different plant species may facilitate efforts to promote the resilience of ecosystems."
I think the authors did not discuss anything on this issue in the rest of the paper, so I think this text is relevant in the Abstract.

REPLY: We agree that the aspect of ecosystem resilience goes too far here, although we point to an ecosystem perspective in the discussion. We rephrased the Abstract accordingly (lines 29).

L 476-477 "However, leaf habit is not necessarily related to physical defenses [52]"
I think this text is not needed as it neither adds much to this paper nor to science. (I quickly checked the cited paper but I got a question whether they compared deciduous leaves and evergreen leaves at the right timing during the season. Also, they made mistakes in both terminology and units.)

REPLY: We agree and deleted this statement in the discussion as suggested (lines 505).